# Perceived power dynamics in nursing education on students' learning experience in Ghana

**Collins Atta Poku**[1], **Veronica Adwoa Agyare** [2]*, **Samuel Baafi**[3],
**Priscilla Yeye Adumoah Attafuah**[4,5], **Eunice Berchie**[6]

1 Department of Nursing, School of Nursing and Midwifery, KNUST, PMB UPO, KNUST, Kumasi, Ghana, 2 Akenten Appiah-Menka University of Skills Training and Entrepreneurial Development (AAMUSTED), Kumasi, Ghana, 3 Gloucestershire Royal Hospital, Gloucester, United Kingdom, 4 School of Nursing and Midwifery, College of Health Sciences, University of Ghana, Accra, Ghana, 5 Geriatric Hub LBG, Madina, Accra, Ghana, 6 Seventh Day Adventist Nursing & Midwifery Training College, Kwadaso, Kumasi, Ghana

All authors contributed equally to the study.

* adwoavagyare@yahoo.com

## Abstract

### Introduction

Educator-student connectedness is where students can thrive, even amid failure, preserving their worth and self-dignity. Every relationship may have challenges, which is no different in nursing education. Students are sometimes unhappy with the educator-student relationship, as power creates a certain distance that must be appropriately exhibited. The study explored the perceived power dynamics of the educator-student relationship in nursing education.

### Methods

The study utilised an exploratory, descriptive qualitative approach to understand student nurses' educator-student interaction. Twenty-four final-year students were involved in the study based on information power. Thematic analysis was used to generate codes, sub-themes, and themes.

### Findings

Two main themes emerged: 1) difficulties faced by students and 2) the perceived impact of power dynamics. The identified subthemes include issues with teaching methods, feelings of suppression, and discouraging attitudes of educators. The findings further showed that students often pretend to understand when an educator becomes angry, reflecting the power dynamics in the classroom, where students may feel powerless to challenge authority even when their academic needs are unmet. Additionally, there is a lack of student autonomy, which affects their mental well-being and development as competent professionals.

**Data availability statement:** All relevant data are within the paper and its Supporting information files.

**Funding:** The author(s) received no specific funding for this work.

**Competing interests:** None of the authors have competing interests.

## Conclusion

Students are expected to develop critical thinking skills and become advocates for their patients; however, when their voices are suppressed, it is unlikely that they will confidently advocate for others in clinical settings. Suppressing their concerns can lead to long-term consequences, affecting their ability to question clinical decisions and their willingness to speak up for patient safety. Educational institutions should provide professional development on creating inclusive and supportive classroom dynamics to mitigate the adverse effects of power imbalances.

## Introduction

Nursing education prepares future nurses to deliver high-quality patient care in healthcare settings. The nurse educator's teaching, supporting, caring, guiding and nurturing role depends on how students perceive the interaction. The relationship can have either a positive or negative impact on learning outcomes, supporting students in developing both personally and professionally and enhancing their self-worth [1].

Several studies have documented the positive effects of the educator-student relationship on students' learning experiences, including support, enhancing self-confidence and self-worth, increasing motivation to learn, and maximising class and clinical learning outcomes [2–4]. The positive nature of educators' attitudes toward students stimulates the essence of putting oneself in a student's situation to enhance learning in the educational setting [5–7]. At the same time, the relationship facilitates strength-based learning and nursing education, while a positive educator-student relationship provides a sense of belongingness [8–10].

From another perspective, educators' constructive feedback helps students identify their strengths and weaknesses, enabling them to improve [11,12]. This phenomenon demonstrates that good relationships lead to improved learning outcomes, including students' attentiveness in class, asking questions, and freely discussing academic issues [13]. Likewise, a poor relationship or interaction will result in students showing their displeasure by absenting themselves from the class or being present in a lecture but not listening or participating in activities [14]. Boat et al. [15] asserted that a poor student-educator relationship is worse than not having one, as it results in strong negative emotions in students. This poor relationship may inevitably lead to students not participating in clinical experiences and even absenting themselves from the ward.

Consequently, the nature of the educator-student relationship, which often manifests in hierarchical structures, can create an environment where power imbalances become pronounced [16,17]. Power typically resides with educators who decide on students' learning experiences and assessments. As a result, students may perceive themselves as vulnerable and dependent on educators for access to learning opportunities, evaluations, and eventual success in their coursework and clinical practice [18,19].

In this context, power dynamics refer to the distribution and exercise of authority, control, and influence within an educational setting, and this phenomenon has

become a crucial issue in nursing education [20,21]. Various factors, such as institutional policies, cultural norms, and individual attitudes, can influence these dynamics. For instance, grading practices, mentorship style, and the perceived openness of the learning environment all contribute to how students experience and navigate power relations. When power imbalances are pronounced, students may feel less autonomous, hesitant to question or challenge ideas, and reluctant to participate actively in learning activities [22].

Studies have indicated that perceived power imbalances in educational settings can negatively influence students' motivation, self-efficacy, and overall educational outcomes. When students feel constrained or intimidated by power dynamics, they may hesitate to seek clarifications, question clinical decisions, or engage in reflective learning behaviours essential for developing critical thinking skills and professional independence [23–25]. This reluctance to participate fully in the learning process can adversely affect short-term performance and long-term professional development, leading to gaps in competency, confidence, and patient care quality [26].

Moreover, the hierarchical nature of healthcare institutions may exacerbate these power differentials. Students often occupy the lowest rank in the clinical hierarchy, being subordinate to educators, preceptors, physicians, senior nurses, and other healthcare professionals [27]. This layered dynamic can further marginalise students, limiting their opportunities to advocate for themselves and fostering a hidden curriculum where passive compliance is tacitly reinforced. Although efforts to promote collaborative, student-centred pedagogies in nursing education have gained momentum [28], the existing literature suggests that perceived power imbalances persist as barriers to effective learning [29].

Despite increased attention to the effects of power relations on the learning experience, gaps remain in understanding the mechanisms through which power imbalances manifest and how these perceptions impact students' academic and clinical expertise [19]. There is also a gap in how students feel that the perceived power imbalance influences their learning experience [30]. In Ghana, the nursing profession is based on hierarchical structures, with rankings ranging from lower to higher grades depending on qualifications and years of experience. Student nurses/midwives are more recognised at the entry points, which indicates that they are the least ranked. By focusing on perceived power imbalance as a distinct phenomenon, stakeholders can better identify strategies to enhance students' autonomy, empower them within the learning process, and ultimately improve learning outcomes. The study, therefore, explored the perceived educator-student power dynamics in nursing education and its influence on the learning experience.

## Theoretical framework

Nursing education often involves hierarchical structures, where educators hold significant decision-making power over students. The notion of a perceived power imbalance in nursing education can have implications for students' learning. These may include fear, silence, or compliance among students, which can impact their academic performance, self-efficacy, and professional identity formation.

The study was designed based on Freire's Critical Pedagogy Theory [31]. The theory provides a lens through which power structures and their effects can be examined meaningfully. It critiques the traditional "banking model" of education, where the teacher is viewed as an all-knowing authority transmitting knowledge to passive student recipients. It positions students as active co-creators of knowledge, emphasising dialogue, critical thinking, and the pursuit of social justice within the educational process [32].

The key tenets of Critical Pedagogy include dialogue and participation – learning is a collaborative process in which educators and students engage in meaningful conversations, challenging existing power relations. Again, reflexivity – where educators and learners reflect on their assumptions, biases, and the power structures that shape the educational process; and empowerment – education should equip students to critically analyse societal and institutional injustices, transforming them into change agents [33,34]. Recognising these contextual distinctions between the Western world and SSA is essential for nursing students in Ghana; while power dynamics may be more pronounced, students' voices remain invaluable. Their active participation is vital not only for their own learning and professional development but also for

advancing a more inclusive and reflective nursing education culture that can ultimately transform clinical practice. However, due to the foregrounding issues of power, control, and student empowerment in nursing education, Critical Pedagogy is well-suited to explore how imbalances in authority affect educational outcomes from the students' perspectives.

## Methods

### Research design

The study employed an exploratory, descriptive qualitative approach [35] to investigate the perceived educator-student power dynamics in nursing education and their impact on the learning experience. This approach is well-suited to understanding how educator-student power dynamics influence students' learning experiences.

### Study setting

The study was conducted at a Nursing Training College in Kumasi. The school educate nurses to attain the requisite confidence and competencies to provide care to clients. Teaching is progressive from the first to the final year, where students are prepared for the licensing exams to become professional nurses. Students recruited into the nursing programme have completed high school, equivalent to grade 11, in high-resource countries.

### Inclusion criteria

The participants in the study included final-year students enrolled in the Diploma in Nursing programme, which focused on general nursing and midwifery. They were recruited into the study because they had experienced varied interactions with educators and would be better positioned to provide in-depth information.

### Sampling and sample size

The study employed purposive sampling to recruit participants who could provide rich, relevant, and diverse perspectives on perceived power imbalance in nursing education. Based on the inclusion criteria, students interested in sharing their experiences were recruited for the semi-structured interview. The interviews were conducted based on the availability of participants and the agreed-upon time. The study included a sample size of 24 participants. The number was guided by information power [36]. The researchers conducted no more interviews since no new information was identified. The researchers ensured coding and meaning saturation to gain a deeper understanding of the study phenomenon.

### Data collection procedure

Permission was sought from the school's heads for the study. Participants were informed by their class leaders and through an announcement. Those who were willing to participate were contacted, and a date and venue were agreed on. Before the interviews, prospective participants read the information sheet and sought clarification before signing the consent form. They readily signed the consent form. The researchers met students after class hours to engage them in a conversation about the study. This strategy prevented any interference with their routine teaching and learning schedule. Recruitment of participants took place from March 7, 2018, to April 13, 2018. The researchers developed a semi-structured interview guide with specific areas to initiate discussion on the topic: 1) Share your views on the educator-student relationship. 2) Kindly share some of the power issues in the educator-student relationship (describe with examples). 3) In your opinion, how do the power issues between educators and students influence students' learning experience? Further clarifications and questions were elicited based on the participants' responses. All the interviews were audio-recorded with participants' permission, and each interview lasted between 30 and 60 minutes. Detailed field notes were taken to capture non-verbal cues and researchers' observations.

## Data analysis

Braun and Clarke's thematic analysis [37] was applied to the data. Through an iterative process, data analysis commenced with transcribing the audio-recorded data after each interview. The researchers took turns verifying the audio with the transcripts to ensure the completeness of the information. Individual researchers read through the transcripts to familiarise themselves with the data. The researchers identified and labelled meaningful data units (codes) across the data set. Similar codes were combined into themes. An analysis was done to define each theme clearly, determine sub-themes, and establish the essence of each main theme. The researchers ultimately produced an integrated analytical narrative using the extracted data.

## Methodological rigour

Rigour was assured through credibility, dependability, confirmability, and transferability [38]. Some strategies instituted were prolonged engagement by providing enough time for participants to answer questions, member checking and debriefing to ensure data credibility. At the same time, an audit trail illustrated all the processes employed during the study, from raw data to textual data and inter-coder checks were done to ensure dependability. To ensure confirmability, the researchers maintained reflexive journaling, bracketed personal assumptions, and ensured that their findings were grounded in the data rather than in researcher bias. All the researchers are educators, with three from the institution of study. The researchers acknowledged their biases, including awareness of power imbalances, hierarchical structures, and students' struggles. Participants were informed of the study's essence, including its benefits and implications for practice. This information created an environment for openly sharing their experiences.

Thick descriptions of the context, participants, phenomenon under study and the interpretations of responses were conducted to enable readers to assess the study's relevance to other contexts (transferability).

## Ethical considerations

Data collection commenced after obtaining ethical clearance from the Committee of Human Research, Publication and Ethics of the Kwame Nkrumah University of Science and Technology (CHRPE/AP/067/18), where the school is an affiliate. Informed consent was obtained from participants, as well as permission from gatekeepers. The study's risks, benefits, and scope were spelt out to participants. Participants were approached individually and in smaller groups after the closing lectures to prevent any potential fear or anxiety about whether to participate or not in the study. Additionally, with permission from the school authorities, the study was announced in the final year of the student's class to create awareness about the research.

Students were then informed to see the researchers after the lectures were closed if they were interested in the study. All potential participants who showed interest were informed that the study was not intended to find faults but rather to enhance students' educational experiences. To reduce power dynamics during the recruitment process, we established rapport and informed them of the need to understand their experiences and challenges in order to enhance students' educational experiences. Additionally, researchers disclosed to participants their awareness of power dynamics between students and nurse educators, but we need evidence to support this. Potential participants verbally volunteered to participate in the study after understanding its purpose. All interviews were conducted either after the lectures closed or on weekends in the demonstration room to ensure that students could openly share their experiences without feeling pressured or having perceived thoughts that others would be observing. Privacy and confidentiality were maintained through the use of pseudonyms during the interviews. The researchers identified a psychologist to counsel students who may remember painful experiences. However, none of the participants requested a psychologist. All transcripts were coded to prevent linking quotes to participants. Only participants who voluntarily showed interest in the study were involved.

### Findings

The study revealed two main themes, each with five associated sub-themes, based on the participants' narratives. The themes were difficulties faced by students and the perceived impact of power dynamics. Table 1 details the themes and related sub-themes from the findings.

### Participants characteristics

A total of 24 participants aged between 21 and 29 years were involved in this study. Participants had completed Senior High School (SHS) and enrolled in nursing education. Eighteen (18) participants were female, and twelve (12) males were in the final year of their education. All participants were Ghanaians who spoke English.

## Theme 1: Difficulties faced by students

These were difficult situations or encounters that students faced during their interactions with educators. The narrations revealed that students experienced several difficulties related to power dynamics between them and educators. The foundation of these power dynamics was linked with issues of teaching. The identified subthemes included issues with teaching methods, feelings of suppression, and a discouraging attitude among educators.

### Issues with teaching methods

They involve the teaching approaches adopted by the educators. Some students claimed that the teaching approaches were usually less suitable for their learning. They revealed that some educators chose a particular teaching method because they wanted to complete the course outline quickly. Others were concerned that educators did not provide clear standards or guidelines for skill acquisition. This situation led them to apply their understanding and criteria, which the educators usually do not accept. Invariably, it creates the notion that the students are not serious during lessons.

> "The tutors share the topics with us in groups to make us complete the syllabus faster. They set questions on all those topics. A group may not understand ideas from the other group, which affects the students during an examination." (P3)

> "During our practical sessions in the skills laboratory at level 200, we were not given the standard component task on what we were supposed to do. If you make a mistake, the comment that the educators will make, such as 'you people,' is that when we were teaching, you learnt nothing from it. It was as if they had provided us with the standard guide, and we had refused to comply. They are still doing that even now." (P14)

A student narrated how he felt a method, though appropriate, later became a nuisance. He said this prevented most students from having enough private time to learn.

> "We are usually grouped for various discussions and presentations. We had some positive experiences from the groupings, but it also affected us in another way. Positively, we tend to participate in group discussions because we are at the

**Table 1. Themes and Subthemes generated from findings.**

| Themes | Sub-themes |
|---|---|
| Difficulties faced by students | a. Issues with teaching methods<br>b. Feelings of suppression<br>c. Discouraging attitude of educators |
| Perceived impact of power dynamics | a. Positive impact<br>b. Negative impact |

*same level of understanding. It brings out the insight that we need as students among ourselves, and we thoroughly enjoyed it. However, there are too many group discussions. Almost every day, every period, we meet as a group. It interrupted our studies, and most students had terrible grades that semester." (P5)*

Concerning the appropriateness of teaching methods, few students mentioned that educators made learning more accessible, while the inappropriate ones led to difficulties in learning. They were enthused and appreciated the appropriate approaches as they enhanced learning. Some teaching methods that students preferred were the utilisation of audio-visual aids, visual presentations, and participatory teaching methods such as demonstrations and return demonstrations. The students emphasised that they could quickly reflect and learn from courses taught using such approaches.

*"Some of the tutors demonstrate and use videos. I like learning about their subjects because I can reflect on what happened in class by comparing them to the reading materials. However, some tutors read the handouts without providing examples. It makes it difficult to understand the concept." (P2)*

*"I remember vividly we were learning something about meningitis, and the tutor called one student to lie on the table and demonstrated the various signs, such as Brudzinski's. It was beneficial. We always remember the signs." (P18)*

## Feelings of suppression

Students emphatically stated that they had no say in the relationship, especially when they were made aware that, as students, they must abide by rules. These circumstances displayed a power struggle between the educators and students. The students were automatically displaced since educators assumed the role of superiors with inherent power and influence in the teaching and learning interactions. Some students complained that they could not comfortably discuss concerns with educators. They felt scared, possibly because it could lead to being labelled as troublesome, which would affect their examination results.

*"We are brought up not to talk about certain issues in the school. You have to obey the tutors. If you go against, you will have a problem. I do not want to be withdrawn or have a problem with anybody. I do not think I would take the risk, considering the money my parents have spent on my education." (P6)*

*"The students are not given the chance to speak their mind on certain issues in the classroom. When you try, you would be in danger." (P17)*

A student felt powerless. He could not question the educators' stance or the administration's decisions, which was frustrating. Invariably, such feelings of suppression could prevent them from being open-minded and curious when caring for patients in the hospital. One of the students shared an experience she will never forget, which concerns their licensing examination.

*"After we realised that we would be writing our licensing exam almost a year after completing school, we went to seek redress from the head of the department, and the response was that we were children and should not be concerned with administrative issues. It is supposed to be confidential, not to be rumoured. As students are being educated to become professionals in the future, if they instil this thought in us, it would not help. If students are aggrieved and have concerns, you should address them. This is an experience, I will not forget." (P15)*

### Discouraging attitude of educators

According to the students, the behaviours of educators discouraged them from learning. They attributed such behaviour to the negative utterances from the educators during any form of interaction. It seemed the educators dwelt on students' negatives and did not employ a strength-based approach in their interaction. This attitude may have generated negative energy among the students, making them reluctant to make room for learning. Students were deeply concerned about this discouraging stance. They could not readily ask questions in class.

*"I think sometimes the attitude of the educators hinders students from asking questions. Sometimes, when you ask questions, the tutor will say, 'I don't have time; I'm going somewhere.' If the educator has such a personality, you cannot ask questions of that educator." (P16)*

Specifically, students were disheartened by comments suggesting they possessed a low intelligence quotient (IQ) level. They disclosed that some students quickly understand concepts more than others. It is, therefore, essential that educators consider the diverse learning styles of students in their lesson preparation and be accommodating of their individual needs to foster a productive educational experience.

*"Sometimes, some educators often discourage us when they come to teach. Our IQ levels are not the same. Some students understand more quickly than others…Comments from some educators, like, "Do I need to teach this topic?", do not help some of us, meaning we are not smart or we have a low IQ. Also, you will not even get the chance to go to them for an explanation if you do not understand a topic". (P13)*

Others reported that they were very displeased with the persistent comparison of their learning abilities with those of the educators when they were also students.

*"Some educators comment on our ability to learn, comparing their generation to our current generation; we have low learning abilities, especially IQ. I was unhappy about it because we are all different". (P2)*

*"An educator told us that we do not like learning at all or are lazy. She insisted that, during their time, they were studying hard. Meanwhile, the number of subjects we study currently exceeds their time." (P9)*

## Theme 2: Perceived impact of power dynamics

The perceived impact of power dynamics is the influence of the interaction on students' learning experience. Students experienced positive and negative impacts, which were identified as subthemes. The narrations depicted a more negative experience than a positive one.

### Positive impact

These were the reflections of how students felt that the interactions with educators were helpful in their learning experiences. Few students reported that when the interaction with educators was good, it stimulated their learning, making them readily participate in class. For educators who exhibit discouraging behaviours, students often want them to leave the class quickly once they have finished teaching.

*"Students are always ready for those who are considerate and have a good relationship with us. They are always ready to listen to and support us in our learning. This relationship is reflected in our exam results, because I aim to impress such educators that their efforts were not in vain". (P23)*

*"If you have a good relationship with an educator, you like to learn what he/she teaches." (P1)*

## Negative impact

Notwithstanding some positive effects, most students felt intimidated to ask questions in class. They reported that the attitude of some educators had a negative impact on them. Students often struggle to pay attention in class, which could negatively impact their academic performance.

*"…When they are teaching, you feel intimidated to ask questions." (P8)*

*"There was a time a lecturer was in class, and I asked a question; the lecturer did not know the answer to give. Instead of telling me that she would probably research and find an answer for me later, she replied by saying, 'Why did you ask that question?' I was embarrassed; since then, I listen to her whenever she comes to class, but nothing comes to mind."* (P22)

A student disclosed that he feigns understanding whenever he asks a question, and the educator becomes angry. This student's approach was to prevent any negative attitude from educators.

*"In the classroom, I ask questions, but immediately I see that the educator is getting angry, I just stop or just say "I am okay, even if I do not understand." (P21)*

Another mentioned that she is experiencing challenges with his grades due to the attitude of some educators.

*"...we do not want some educators to come to class. They start insulting and giving us negative words before they begin to lecture. After the lecture, we would all say no when the tutor asked us questions because we wanted her to leave the class. I realised it is affecting my grades, and I'm worried." (P12)*

Apart from the difficulties and perceived impact, students suggested recommendations to improve educator-student interactions and teaching and learning. Some students mentioned that students should be encouraged to raise concerns for redress rather than obeying rules without complaining. Again, educators should create a positive learning environment that reflects on students' learning.

*"We understand that the educators are our superiors, and we need to respect them, but they need to listen to our concerns and be more accommodating". (P7)*

*"We need more encouragement from the educators. It is very helpful." (P2)*

Some students suggested that incentives for educators and students would stimulate positive interaction. Concerning students' incentives, it would motivate and encourage independent learning. They mentioned that incentives could be in the form of marks or money.

*"I think we should budget incentives for students so it will motivate us to speak out in class when necessary. I recall that in our communication skills class, the educator would sometimes ask a question, and when you tried to answer, he would give you a mark of +1 or +5. Those instances made us more attentive to what she was teaching. We were also active in class, answering and asking questions so that at least we would get some marks to supplement what we got in our assignments and classwork" (P24).*

*"I think the incentive is very vital. One educator usually does that; she will ask a question, and when you get it right, she gives you money because she knows you have researched it, and it makes students very active and readily research a specific topic. Students prepare before the educators come in because there is something for you"* (P8).

Another participant suggested that the institution should reward academically outstanding students and those who have shown improvement to encourage others to do the same during ceremonies.

*"We have been awarding first-class students, but some students were not doing well; their first GPA was 1. If such a student improves tremendously in the second-year semester, we must reward them to boost their morale and encourage them to continue learning. This will allow others to learn from it as well. It should not be only first class"* (P6).

A participant noted that educators' incentives will motivate them, especially considering the numerous students for whom they are responsible.

*"I suggest we motivate our educators. They do not get any motivation, and that is why they insult us."* (P14)

Also, other students suggested the use of the local language in teaching

*"When the educators are teaching, they should use both the English language and the local dialect to help some of us to get the understanding."* (P11)

*"Educators must use simple expressions in teaching the students for us to understand."* (P18)

## Discussion

Educators have a significant influence on students' learning experiences, particularly in practice-based disciplines where skill acquisition is crucial. The study examined the perceived educator-student power dynamics in nursing education and their impact on students' learning experiences. There is a growth of enthusiasm for teaching approaches or methods that students deem appropriate, particularly those that enhance their understanding through practical application and visual aids. The findings highlighted the need for teaching methods such as demonstrations, audio-visual materials, and participatory learning methods, including return demonstrations, rather than group presentations. Educational researchers believe these methods align with adult learning theories, which emphasise the need for active learning, reflection, and the application of knowledge to real-world contexts. Therefore, they must be promoted for use in higher learning environments [39,40]. Educators must consider the varied learners and utilise approaches that enhance their learning rather than adopting a stance that is skewed towards a particular approach that challenges students' learning.

Students' feelings of being powerless and silenced reflect a broader issue in educational systems, where educators often assume a position of unchecked authority [21,41]. In Freire's theory of Pedagogy of the Oppressed, the hierarchical relationship between educators and students stifles critical thinking and open dialogue [31]. Freire's position agrees with the findings from this study, which showed that students are not allowed to voice their concerns, as questioning authority could result in being "tagged as troublesome." Students fear that speaking out would lead to punitive measures, further cementing their subjugation within the educational hierarchy. This power imbalance in nursing education is often justified under the guise of maintaining order or upholding institutional norms, but the implications of such a dynamic are significant [42]. Moreover, when students are not given a platform to express their grievances, it suppresses their critical thinking and autonomy, essential attributes in professional healthcare practice [43,44].

As shown in the study findings, some students reiterated that when educators maintained positive relationships with students by respecting and valuing them, it enhanced their learning and classroom participation. This finding is consistent with a supportive learning environment, which suggests that positive student-educator interactions foster a sense of

belonging, stimulate active engagement and motivate students to learn and participate actively in the learning process [45]. When educators create a respectful and open classroom environment, students are more willing to take risks, such as asking questions and engaging in discussions, which promotes deeper learning [46,47]. Therefore, mutual respect is essential to streamline power and create an atmosphere where students feel safe and free to communicate [48].

Conversely, the study findings showed the negative impact of discouraging educator behaviours, such as insults or dismissive attitudes, which led to disengagement and avoidance of class participation. This finding echoes previous research that emphasises the detrimental effects of educator intimidation on student learning [49]. When students feel threatened or humiliated, it creates an emotional barrier that hinders cognitive engagement and learning retention [50]. These circumstances usually damage students' trust in their educators and negatively affect their academic performance [51–53].

Additionally, the findings from this study revealed an overarching experience of suppression of learning among students, attributed to the attitudes and behaviours of educators, which hindered their capacity to engage in meaningful educational experiences, as educators seemed to focus on students' weaknesses rather than adopting a strengths-based approach. This finding aligns with Aldrup et al. [54], highlighting the negative impacts of teacher-student interactions that lack empathy and support can have on suppressing student motivation and academic engagement. Strengthen-based approach dwells on the positives to enhance the negatives. It encourages students to press on to achieve the best outcomes by building on what works best [55]. Educators must learn to use such approaches, gradually building students' confidence.

Moreover, the study's findings indicated that students' inability to ask questions in class was due to educators' dismissive behaviour, and it reflects a significant barrier to learning, suggesting that open communication between students and educators is essential for educational success [56]. This finding reiterates a similar position of other studies [57,58], which argue that creating an atmosphere where students feel comfortable voicing questions is fundamental to a positive learning experience. Furthermore, the negative comments about students' intelligence levels highlight another layer of suppression [59,60]. On the concept of "fixed" versus "growth" mindsets, studies support the notion that when educators label students as inherently less intelligent or capable, it stifles their ability to develop and improve over time [32,61]. Students' self-perception, particularly in response to feedback from authority figures, plays a significant role in their academic identity and achievement [62,63]. Therefore, educators' comments diminish the students' sense of agency and reinforce a fixed mindset, where they begin to believe that their intelligence cannot change. These findings point to the importance of professional development for educators, particularly in understanding the impact of their language and behaviour on students' learning experiences. Reflective teaching practices, which include self-awareness of one's biases and the adoption of diverse instructional approaches, are essential in creating a supportive educational setting that empowers rather than suppresses student learning [64].

Again, the psychological impact of suppressing student voices is profound, as evidenced by the findings of this study. Some students expressed a sense of danger if they tried to speak their minds, revealing a fear of retribution. The findings further showed that students often pretend to understand when an educator becomes angry, reflecting the power dynamics in the classroom, where students may feel powerless to challenge authority even when their academic needs are unmet. This finding is consistent with studies by Allen et al. and Scherr and Mayer [65,66], which consistently show that environments where students feel marginalised or unable to express their opinions contribute to anxiety, decreased self-esteem, and academic disengagement. This environment fosters a culture of silence, discouraging students from challenging the status quo and thereby limiting their educational growth.

Moreover, a lack of student autonomy affects their mental well-being and development as competent professionals. Students are expected to develop critical thinking skills and become advocates for their patients; however, when their voices are suppressed, it is unlikely that they will confidently advocate for others in clinical settings [67,68]. Suppressing their concerns can lead to long-term consequences, affecting their ability to question clinical decisions and their willingness to speak up for patient safety.

## Implications for professional practice

The core competency in nursing and healthcare education is effective communication, which is essential for ensuring patient safety, promoting interprofessional collaboration, and resolving ethical dilemmas. If students are discouraged from expressing concerns during their education, they may carry this reticence into their professional practice, potentially compromising patient care or other aspects of their work. To mitigate feelings of suppression and promote a healthier learning environment, educators must adopt a more participatory approach that actively engages students in their learning. This approach creates spaces where students can safely express their opinions and concerns without fear of retribution.

The findings directly link classroom dynamics to critical outcomes, such as student autonomy, mental well-being, and professional competence, offering actionable insights for improving educational policy and practice. Educators should, therefore, not be seen as authoritarian figures but as facilitators who guide students in developing their critical thinking and professional skills. By doing so, educators can help dismantle the oppressive power structures prevalent in many educational institutions. Again, educators must create a psychologically safe classroom environment where students feel comfortable asking questions without fear of judgment. Academic institutions should provide professional development on creating inclusive and supportive classroom dynamics to mitigate the adverse effects of power imbalances.

## Limitations and future research

The study presents potential limitations. A relatively small sample size, drawn through purposive sampling, may limit the transferability of the findings to similar contexts. Again, there was a potential for researcher bias during data collection and analysis. The study employed trustworthiness strategies, such as peer debriefing, member checking, and consensus-building discussions, to mitigate the limitations. Despite these limitations, the study employed a rich qualitative approach that allowed for a nuanced understanding of the complex power dynamics between educators and student nurses. Again, the inclusion of 24 students guided by information power ensured that the findings were grounded in sufficiently rich and diverse perspectives, enhancing the credibility of the results. Future studies will benefit from using a mixed-methods design, which enables researchers to triangulate findings and improve the generalisability of the results. A longitudinal study that follows students over time can provide a deeper understanding of how perceived power imbalances evolve and influence learning experiences throughout their nursing education.

## Acknowledgments

The authors acknowledge that all participants were used in this study.

## Author contributions

**Conceptualization:** Collins Atta Poku, Veronica Adwoa Agyare, Samuel Baafi, Priscilla Yeye Adumoah Attafuah, Eunice Berchie.

**Data curation:** Collins Atta Poku, Veronica Adwoa Agyare, Samuel Baafi, Priscilla Yeye Adumoah Attafuah, Eunice Berchie.

**Formal analysis:** Veronica Adwoa Agyare, Samuel Baafi, Priscilla Yeye Adumoah Attafuah.

**Investigation:** Veronica Adwoa Agyare, Samuel Baafi.

**Methodology:** Veronica Adwoa Agyare, Eunice Berchie.

**Supervision:** Collins Atta Poku.

**Writing – original draft:** Collins Atta Poku, Veronica Adwoa Agyare, Samuel Baafi, Priscilla Yeye Adumoah Attafuah, Eunice Berchie.

**Writing – review & editing:** Collins Atta Poku, Veronica Adwoa Agyare, Samuel Baafi, Priscilla Yeye Adumoah Attafuah, Eunice Berchie.

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
