## [Decision Letter · Decision Letter 0]

Dear Dr. Agyare,

Thank you for submitting your manuscript to PLOS ONE. After careful consideration, we feel that it has merit but does not fully meet PLOS ONE’s publication criteria as it currently stands. Therefore, we invite you to submit a revised version of the manuscript that addresses the points raised during the review process.

We look forward to receiving your revised manuscript.

Kind regards,

Paridhi Jha, PhD

Academic Editor

PLOS ONE

Journal Requirements:

3. We note that your Data Availability Statement is currently as follows: Data are all within the manuscript

Reviewers' comments:

Reviewer's Responses to Questions

**Comments to the Author**

1. Is the manuscript technically sound, and do the data support the conclusions?

Reviewer #1: Yes

Reviewer #2: Yes

2. Has the statistical analysis been performed appropriately and rigorously?

Reviewer #1: N/A

Reviewer #2: No

3. Have the authors made all data underlying the findings in their manuscript fully available?

Reviewer #1: Yes

Reviewer #2: No

4. Is the manuscript presented in an intelligible fashion and written in standard English?

Reviewer #1: Yes

Reviewer #2: Yes

Reviewer #1: Title: I suggest your title should include Ghana

Introduction:

It is an important study. Thank you for taking the time to undertake this study.

Methods:

The most current text from Braun and Clarke prescribes a departure from “data saturation” to “information power.” Please look at p.28 of Braun and Clarke (2022).

In line 152, you used “train nurse … training college…” but in line 234, you used “nursing education”. Your title includes nursing education. These phrases are contradictory. You cannot train nurses and educate them simultaneously. I suggest removing train/training from your work even though the students are enrolled in a diploma; the college is affiliated with a university that provides education, not training.

Theoretical framework:

The nursing profession is taught based on set principles and concepts. It would be helpful for readers, especially nursing students, to note that educators can be co-creators of knowledge, but there is a limit to this expectation. However, this concept may not be practicable in Ghana, but Global North countries have strict accreditation to ensure that the contents of accredited nursing degrees are taught as set out in the approved accreditation provisions. Competency standards are essential in nursing education and accreditation. On the other hand, educator-student relationships in the Global North countries significantly differ from their counterparts in the Global South countries like Ghana, Zimbabwe, Nigeria, etc. Educators in the Global South countries are viewed as small gods by students, and it is an entirely different situation in the Global North countries like the UK, the US, Canada, Australia, etc. Please note that I do not discount power differentials between students and educators, but the powers are far less significant in the West.

A paragraph should tease out these points so that nursing students would not feel devalued in their contributions in the classroom.

Ethics:

Study settings and sampling. How did you recruit the participants? Were the participants compelled to participate in the study? What information did you provide to the participants? How did you manage power dynamics in the recruitment process as researchers? Did you conduct the interview on college premises or at a location convenient to the participants?

You invited participants who “voluntarily showed interest” How did you determine their voluntariness?

Findings:

The findings are quite worrying and traumatic to me reading them. I encourage your team to present your conclusions at Ghanaian conferences or other strategic audiences if they have not already been done.

Discussion:

Critical thinking skills need to be embedded in nursing education, and participatory/collaborative teaching promotes these skills. Yet still upholding safety, quality and competency standards that come with nursing accreditation or registrations.

Reviewer #2: The manuscript is excellent, addressing an important topic with a robust methodology and exhibiting excellent drafting.

Please present the main findings in tables or charts for better understanding.

"Given that the data was collected from March 7th, 2018, to April 13th, 2018, why has the article been written now?"

The final paragraph of the "Methodological Rigour" section should be written at the end of the article under the heading: "Authors' Contribution."

Please highlight the strengths of your study.

Please write the results section as a separate section.

It is suggested that references published before 2015 be removed and replaced with newer ones.

**Do you want your identity to be public for this peer review?** For information about this choice, including consent withdrawal, please see our Privacy Policy

Reviewer #1: **Yes: ** Adeniyi Olanrewaju Adeleye

Reviewer #2: No

---

## [Author Response · Author response to Decision Letter 1]

31 May 2025

22nd May 2025

Ref:

Title: “Perceived power dynamics in nursing education on students’ learning experience in Ghana”

Journal:

Dear Dr Jha,

We are very grateful to you and the reviewers for the opportunity to revise our manuscript to improve it. Please kindly find the response below. Thank you!

SN Reviewers Comments Responses

Reviewer #1:

1 Title: I suggest your title should include Ghana ‘Ghana’ has been added to the title on page 1.

2 Methods:

The most current text from Braun and Clarke prescribes a departure from “data saturation” to “information power.” Please look at p.28 of Braun and Clarke (2022). Thank you for the information. The method section has been revised on pages 2 and 8.

3 In line 152, you used “train nurse … training college…” but in line 234, you used “nursing education”. Your title includes nursing education. These phrases are contradictory. You cannot train nurses and educate them simultaneously. I suggest removing train/training from your work even though the students are enrolled in a diploma; the college is affiliated with a university that provides education, not training. This point is well noted and has been revised throughout the manuscript as recommended.

4 Theoretical framework:

The nursing profession is taught based on set principles and concepts. It would be helpful for readers, especially nursing students, to note that educators can be co-creators of knowledge, but there is a limit to this expectation. However, this concept may not be practicable in Ghana, but Global North countries have strict accreditation to ensure that the contents of accredited nursing degrees are taught as set out in the approved accreditation provisions. Competency standards are essential in nursing education and accreditation. On the other hand, educator-student relationships in the Global North countries significantly differ from those of their counterparts in the Global South countries like Ghana, Zimbabwe, Nigeria, etc. Educators in the Global South countries are viewed as small gods by students, and it is an entirely different situation in the Global North countries like the UK, the US, Canada, Australia, etc. Please note that I do not discount power differentials between students and educators, but the powers are far less significant in the West.

A paragraph should tease out these points so that nursing students would not feel devalued in their contributions in the classroom. The write-up on the theoretical framework has been revised as recommended, page 7.

5 Ethics:

Study settings and sampling. How did you recruit the participants? Were the participants compelled to participate in the study? What information did you provide to the participants? How did you manage power dynamics in the recruitment process as a researchers? Did you conduct the interview on college premises or at a location convenient to the participants?

You invited participants who “voluntarily showed interest” How did you determine their voluntariness? All queries have been answered on page 10.

6 Findings:

The findings are quite worrying and traumatic to me reading them. I encourage your team to present your conclusions at Ghanaian conferences or other strategic audiences if they have not already been done.

Thank you.

Well noted.

7 Discussion:

Critical thinking skills need to be embedded in nursing education, and participatory/ collaborative teaching promotes these skills while upholding the safety, quality, and competency standards that come with nursing accreditation or registrations. Thank you. It is significant for professional growth and development.

Reviewer #2:

1 The manuscript is excellent, addressing an important topic with a robust methodology and exhibiting excellent drafting.

Please present the main findings in tables or charts for better understanding. Findings have been presented in Tabular format on page 12.

2 "Given that the data was collected from 7th March, 2018, to 13th April, 2018, why has the article been written now?" The authors had few personal challenges until recently, when they were able to assemble and complete the manuscript.

Thank you.

3 The final paragraph of the "Methodological Rigour" section should be written at the end of the article under the heading: "Authors' Contribution." The final section of the "Methodological Rigour" section has been removed and written under the heading: "Authors' Contribution", page 25.

4 Please highlight the strengths of your study. Thank you.

The strength of the study has been highlighted on page 24.

5 Please write the results section as a separate section. The findings of the study have been separate from the discussion.

6 It is suggested that references published before 2015 be removed and replaced with newer ones. The references have been updated as recommended.

Thank you.

---

## [Decision Letter · Decision Letter 1]

Perceived Power Dynamics in Nursing Education on Students’ Learning Experience in Ghana

PONE-D-25-13650R1

Dear Dr. Agyare,

We’re pleased to inform you that your manuscript has been judged scientifically suitable for publication and will be formally accepted for publication once it meets all outstanding technical requirements.

Kind regards,

Paridhi Jha, PhD

Academic Editor

PLOS ONE

Additional Editor Comments (optional):

Reviewers' comments:

Reviewer's Responses to Questions

**Comments to the Author**

Reviewer #1: All comments have been addressed

Reviewer #2: (No Response)

2. Is the manuscript technically sound, and do the data support the conclusions?

Reviewer #1: Yes

Reviewer #2: Yes

3. Has the statistical analysis been performed appropriately and rigorously?

Reviewer #1: N/A

Reviewer #2: Yes

4. Have the authors made all data underlying the findings in their manuscript fully available?

Reviewer #1: Yes

Reviewer #2: Yes

5. Is the manuscript presented in an intelligible fashion and written in standard English?

Reviewer #1: Yes

Reviewer #2: Yes

Reviewer #1: Excellent manuscript and thank you for undertaking this study. I am pleased you accepted my suggestions. Thank you.

Reviewer #2: The manuscript is excellent, addressing an important topic with a robust methodology and exhibiting excellent drafting.

Thank you. all my comments addressed.

**Do you want your identity to be public for this peer review?** For information about this choice, including consent withdrawal, please see our Privacy Policy

Reviewer #1: **Yes: ** Adeniyi Olanrewaju Adeleye

Reviewer #2: No

---

## [Editor Report · Acceptance letter]

PONE-D-25-13650R1

PLOS ONE

Dear Dr. Agyare,

I'm pleased to inform you that your manuscript has been deemed suitable for publication in PLOS ONE. Congratulations! Your manuscript is now being handed over to our production team.

Kind regards,

on behalf of

Dr. Paridhi Jha

Academic Editor

PLOS ONE